bioengineering/molecular biology

transporter, membrane protein, sucrose metabolism, microbial engineering, butanol, *Clostridium beijerinckii*

**Authors for correspondence:**
Ribo Huang
e-mail: guruace@163.com
Hao Pang
e-mail: panghouse@126.com

# Enhanced sucrose fermentation by introduction of heterologous sucrose transporter and invertase into *Clostridium beijerinckii* for acetone–butanol–ethanol production

Lihua Lin[1,2,†], Zhikai Zhang[1,†], Hongchi Tang[1,2], Yuan Guo[2], Bingqing Zhou[2], Yi Liu[2], Ribo Huang[1], Liqin Du[1] and Hao Pang[2]

[1]State Key Laboratory for Conservation and Utilization of Subtropical Agro-bioresources, Guangxi Research Center for Microbial and Enzymatic Technology, College of Life Science and Technology, Guangxi University, Daxue Road No. 100, Nanning, Guangxi 530005, People's Republic of China
[2]Guangxi Key Laboratory of Bio-refinery, National Engineering Research Center for Non-Food Biorefinery, State Key Laboratory of Non-Food Biomass and Enzyme Technology, Guangxi Academy of Sciences, Daling Road No. 98, Nanning, Guangxi 530007, People's Republic of China

(iD) LL, 0000-0001-9269-1707

A heterologous pathway for sucrose transport and metabolism was introduced into *Clostridium beijerinckii* to improve sucrose use for *n*-butanol production. The combined expression of *StSUT1*, encoding a sucrose transporter from potato (*Solanum tuberosum*), and *SUC2*, encoding a sucrose invertase from *Saccharomyces cerevisiae*, remarkably enhanced *n*-butanol production. With sucrose, sugarcane molasses and sugarcane juice as substrates, the *C. beijerinckii* strain harbouring *StSUT1* and *SUC2* increased acetone–butanol–ethanol production by 38.7%, 22.3% and 52.8%, respectively, compared with the wild-type strain. This is the first report to demonstrate enhanced sucrose fermentation due to the heterologous expression of a sucrose transporter and invertase in *Clostridium*. The metabolic engineering strategy used in this study can be widely applied in other microorganisms to enhance the production of high-value compounds from sucrose-based biomass.

†These authors contributed equally to this study.

# 1. Introduction

*n*-Butanol is a valuable chemical used as a solvent and intermediate in a variety of industries [1,2], and is an advanced biofuel alternative to fossil fuels [3,4]. A large amount of butanol is produced each year by chemical synthesis routes with significant environmental impact [5,6]. However, with increasing product demand and environmental consciousness, fermentative methods of *n*-butanol production have gained popularity in recent years. *Clostridium* species are the traditional hosts for butanol production through acetone–butanol–ethanol (ABE) fermentation [7–9]. Unfortunately, many factors limit the capacity of wild-type *Clostridium* strains for butanol production, including low substrate conversion rate, low solvent tolerance, low cell biomass, and excessive byproduct production [10–12]. To overcome these problems, various metabolic engineering strategies have been applied to increase *n*-butanol production in different host strains [11,13–15].

Recently, sucrose-based biomass, a renewable resource, has been increasingly used to produce biofuels and biochemicals [7,16–18]. The native sucrose transport and metabolism pathway in *Clostridium beijerinckii* is a phosphotransferase system (PTS) [19]. Sucrose is first converted into sucrose-6-P via the PTS-dependent sucrose transport pathway, and then decomposed into glucose-6-P and fructose under catalysis by sucrose-6-P hydrolase (figure 1*a*). The product of the gene *scrB* also has the ability to hydrolyse sucrose [20]. However, *C. beijerinckii* lacks an effective sucrose transport system to directly transport sucrose into cells; therefore, the sucrose hydrolase encoded by *scrB* does not play a direct role in sucrose conversion. If a heterologous sucrose transport system was introduced into *C. beijerinckii*, sucrose could be directly taken up into cells and then hydrolysed by ScrB, which could increase ABE fermentation performance using sucrose as the carbon source (figure 1*b*). In a previous study, a sucrose permease from *Escherichia coli* and a sucrose phosphorylase from *Bifidobacterium adolescentis* were co-expressed in *Bacillus amyloliquefaciens*, leading to high poly-γ-glutamic acid production from sucrose [21]. Similarly, enhanced 2,3-butanediol production was achieved in *B. subtilis* by introducing an energy-conserving sucrose usage pathway that combined a sucrose permease from *E. coli* and a sucrose phosphorylase from *Streptococcus mutansa* [21]. Further, Zhang *et al*. achieved high sucrose consumption and ABE production compared with the wild-type strain by deleting a transcriptional repressor gene and overexpressing the endogenous sucrose catabolism pathway in *C. saccharoperbutylacetonicum* N1–4 [7].

A sucrose transporter from potato (*Solanum tuberosum*) was previously introduced into a *Clostridium* strain, which transported a sucrose analogue into cells [22]. The aim of the present study was to explore whether this heterologous sucrose transporter could induce a sucrose transport function, thereby enhancing sucrose fermentation by *Clostridium* for *n*-butanol production.

# 2. Material and methods

## 2.1. Microorganisms and cultivation conditions

The strains and plasmids used in this study are listed in table 1. *Escherichia coli* strain JM109 (Takara Bio Inc., Dalian, China) was used for routine DNA cloning and was cultivated in Luria–Bertani (LB) broth or on LB agar supplemented with antibiotics as needed. *Clostridium beijerinckii* strain 13-4 and derived strains were cultivated anaerobically at 30°C in tryptone yeast extract acetate (TYA) medium [22] or on TYA agar, and 50 µg ml$^{-1}$ erythromycin was added to the medium when required.

*Clostridium beijerinckii* strain 13-4 was isolated from samples collected from soil soaked with cow dung from a buffalo farm (Nanning, China). Soil was diluted and spread on a TYA agar plate containing glucose 40 g l$^{-1}$, yeast extract 2 g l$^{-1}$, tryptone 6 g l$^{-1}$, beef extract 2 g l$^{-1}$, $NH_4Ac$ 3 g l$^{-1}$, $K_2HPO_4$ 0.5 g l$^{-1}$, $MgSO_4$ 0.2 g l$^{-1}$, $FeSO_4 \cdot 7H_2O$ 0.01 g l$^{-1}$ and agar 15 g l$^{-1}$. The plates were incubated under anaerobic environment at 37°C. The clones were picked and sub-cultured to test tube containing TYA medium. Then, the gas-producing clones were selected from the test tube for ABE product analysis by gas chromatography (GC). Further, the ABE-producing clones were selected and sub-cultured to molasses screening medium containing molasses 120 g l$^{-1}$, yeast extract 5 g l$^{-1}$, $Ca(H_2PO_4)_2 \cdot H_2O$ 0.7 g l$^{-1}$, $(NH_4)_2SO_4$ 3 g l$^{-1}$, $CaCO_3$ 3 g l$^{-1}$, NaCl 1 g l$^{-1}$ and $MgSO_4$ 0.4 g l$^{-1}$. Finally, through the comparison of ABE yield, the suitable strain for fermentation on the medium was screened.

## 2.2. Construction of recombinant strains

To enhance sucrose consumption, the sucrose invertase gene *SUC2* (NCBI gene ID: 854644) from *Saccharomyces cerevisiae* was first cloned into the pSOS95-StSUT1 plasmid. *StSUT1* and *SUC2* were

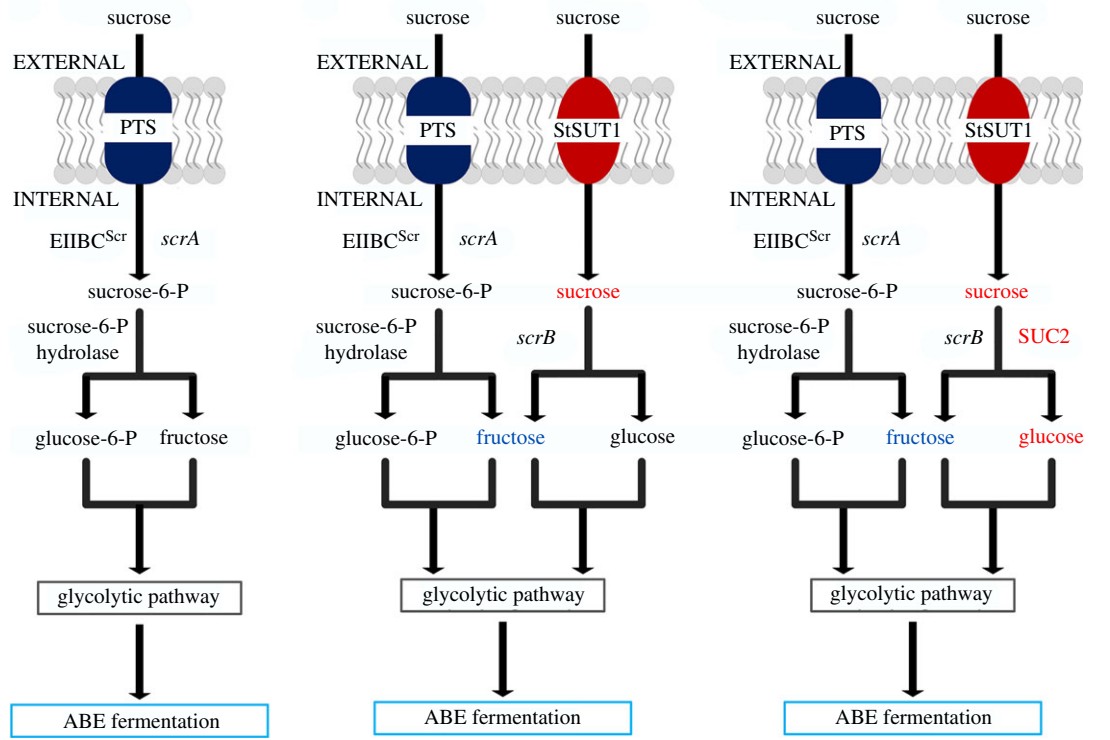

**Figure 1.** Schematic of different strategies for sucrose transport and metabolism in *Clostridium beijerinckii*. (*a*) The native sucrose transport and metabolism pathway consists of a phosphotransferase system (PTS) and sucrose-6-P hydrolase; (*b*) an engineered sucrose transport and metabolism pathway consisting of the native system and sucrose transporter *StSUT1* from potato; (*c*) an engineered sucrose transport and metabolism pathway consisting of the native system, sucrose transporter *StSUT1*, and invertase enzyme *SUC2* from *Saccharomyces cerevisiae*. Text in red represents heterologous sucrose transport and metabolism pathways. ABE, acetone–butanol–ethanol.

**Table 1.** Strains and plasmids used in this study.

| strains and plasmids | relevant features | source |
|---|---|---|
| *E. coli* JM109 | cloning host strain | Takara Bio |
| *C. beijerinckii* 13-4 | wild-type, isolated by our laboratory | laboratory stock |
| JM109-PN95 | *E. coli* JM109/pAN1/pSOS95 | this study |
| JM109-PN95ST | *E. coli* JM109/pAN1/pSOS95-*StSUT1* | this study |
| JM109-PN95STS | *E. coli* JM109/pAN1/pSOS95-*StSUT1-suc2* | this study |
| CB1341 | *C. beijerinckii* 13-4/pSOS95 | this study |
| CB1342 | *C. beijerinckii* 13-4/pSOS95-*StSUT1* | this study |
| CB1343 | *C. beijerinckii* 13-4/pSOS95-*StSUT1-suc2* | this study |
| pAN1 | methylation modification plasmid | [23] |
| pSOS95 | *Clostridium beijerinckii* expression vector | [24] |
| pSOS95-StSUT1 | expressing target gene *StSUT1* | [22] |
| pSOS95-StSUT1-suc2 | co-expressing target gene *StSUT1* and *SUC2* | this study |

constructed as a polycistron under the thiolase promoter of the pSOS95 plasmid. There is no codon optimization for either gene. The *SUC2* open reading frame was obtained by PCR using primers SUC2-F and SUC2-R, whereas the linearized vector containing the sucrose transporter gene *StSUT1* (NCBI gene ID: 102594012) open reading frame was obtained by PCR using primers pSOS95-StSUT1-F and pSOS95-StSUT1-R (table 2). The NCBI gene ID of vector pSOS95 is AY187686.1. In-Fusion HD Cloning kits

**Table 2.** Primers used in this study.

| primer name | primer sequence |
| --- | --- |
| pSOS95-StSUT1-F | TAAAAATAAGAGTTACCTTAAATG |
| pSOS95-StSUT1-R | CCCTCCTTTATTTAATGGAAAGCCCCATGGCGACTG |
| SUC2-F | GGCTTTCCATTAAATAAAGGAGGGATTAAAATGACAAACGAAACTAGCGATA |
| SUC2-R | CCATTTAAGGTAACTCTTATTTTTACTATTTTACTTCCCTTACTTG |

(Takara Bio) were used for fusion ligation of DNA fragments, and the final recombinant expression plasmid was designated pSOS95-StSUT1-suc2. The genetic annotation and structure of the expression vector pSO95-STSUT1-SUC2 are available within the Zenodo (https://zenodo.org/record/5502692) [25]. Plasmids were introduced into *E. coli* strain JM109-pAN1 for methylation before bacterial transformation into *C. beijerinckii* strains according to a previously described method [23,26].

## 2.3. Sucrose transport and enzyme activity assays

To determine sucrose transporter activity, strains expressing *StSUT1* were grown to an $OD_{600}$ nm of 1.5–3.0 in TYA selective medium, and then the cells were collected and washed three times with phosphate buffer (25 mM $Na_2HPO_4$, pH 5.0). *StSUT*1 activity was determined using a previously described method [22], employing a microplate reader to measure esculin fluorescence (367 nm excitation, 454 nm emission).

To measure *SUC2* activity, cells from the acidogenesis phase (24 h) and solventogenesis phase (48 h) were harvested by centrifugation at $12\,000g$ for 10 min at 4°C. The cells were washed twice and resuspended in citrate-phosphate buffer (pH 6.0). The cells were disrupted using a sonication device (5 s sonication at 250 W for 3 s intervals) to obtain cell lysates. The lysate was then centrifuged at $12\,000g$ for 20 min at 4°C, and the supernatant was collected to measure enzyme activity. *SUC2* activity was measured according to a previously described method [27]. Data are reported as the mean ± standard deviation.

## 2.4. Fermentation

A sucrose medium was used as the ABE fermentation medium, containing 40 g $l^{-1}$ sucrose, 7 g $l^{-1}$ yeast extract, 3 g $l^{-1}$ $NH_4Ac$, 0.5 g $l^{-1}$ $MgSO_4$, 1 g l NaCl and 0.5 g $l^{-1}$ $K_2HPO_4$. Sugarcane molasses (containing 310 g $l^{-1}$ sucrose, 52 g $l^{-1}$ glucose and 49 g $l^{-1}$ fructose) or sugarcane juice (containing 100 g l sucrose, 29 g $l^{-1}$ glucose and 13 g $l^{-1}$ fructose) was diluted and used to replace sucrose in the ABE fermentation medium. Sugarcane molasses was pretreated using a previously described method [28]. Fermentations were maintained at 30°C without shaking for 96 h. All fermentations were performed in duplicate.

## 2.5. Analytical methods

Fermentation samples were taken every 12 h for analysis. Cell density ($OD_{600}$ nm) was quantified using a NanoDrop 2000 spectrophotometer (Thermo Fisher Scientific, Waltham, MA, USA). Concentrations of acetone, butanol and ethanol in the fermentation broth were determined by GC (Agilent 7820 series, Agilent Technologies, Santa Clara, CA, USA). The GC conditions were: flame ionization detector temperature 300°C; ZebronZB-WAX column (Phenomenex, Torrance, CA, USA); forward sample temperature 250°C; column temperature 80°C for 0.5 min, then heated to 220°C at 25°C $min^{-1}$; split ratio 10 : 1; hydrogen flow 30 ml $min^{-1}$; air flow 300 ml $min^{-1}$; nitrogen carrier gas and 1% n-propanol as internal standard. Concentrations of residual sugars (glucose, fructose and sucrose) were analysed using high-performance liquid chromatography (HPLC; Waters 1525/2414, Waters Corporation, Milford, MA, USA) with the following conditions: differential refractive index detector; Alltima Amino column (4.6 × 250 mm, 5 μm; Hichrom Ltd., Reading, UK); acetonitrile : water (75 : 25 v/v) mobile phase; flow rate 1 ml $min^{-1}$; column temperature 35°C.

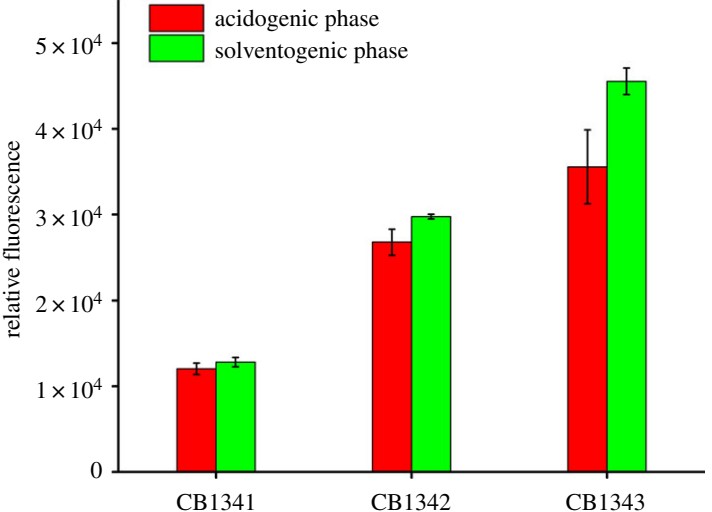

**Figure 2.** Analysis of sucrose transporter activity in *C. beijerinckii* expressing *StSUT1*. CB1341, control strain; CB1342 expressed *StSUT1*; strain CB1343 co-expressed *StSUT1* and *SUC2*.

# 3. Results and discussion

## 3.1. Heterologous expression of *StSUT1* and *SUC2* in *Clostridium beijerinckii*

The *StSUT1* gene, encoding a sucrose transporter from potato, was expressed in *C. beijerinckii*. The sucrose transport activity in the engineered strains was assayed using a previously described method [22], in which highly fluorescent esculin, a structural analogue of sucrose, was used as a probe to characterize sucrose transport activity [29,30]. Strain 13-4 was wild-type strain screened from molasses medium. This strain can be grown and fermented on a simple medium, so it is more suitable for industrial application. Here, the strain is modified to improve its ability to use sucrose. Engineered strains CB1342 and CB1343 exhibited higher fluorescence than the control strain in the esculin transport assay (figure 2). High transport activity of *StSUT1* was observed in both the acidogenesis and solventogenesis phases. These data suggest that *StSUT1* induces efficient sucrose transport in *C. beijerinckii.* The control strain CB1341 exhibited weak fluorescence in the assay (figure 2), indicating that the native sucrose transport system (PTS) of *C. beijerinckii* takes up some esculin. However, the data demonstrate that heterologous transporter *StSUT1* is expressed and functionally active in *C. beijerinckii*.

In native sucrose metabolism in *C. beijerinckii*, sucrose is hydrolysed to glucose-6-P and fructose by a PTS system [19,20]. A sucrose hydrolase (*ScrB*) was also characterized in *C. beijerinckii* [20], releasing glucose and fructose. However, glucose products from sucrose hydrolysis were undetectable in control strain CB1341 or strain CB1342, suggesting that the native sucrose hydrolase activity of *C. beijerinckii* is low and limits effective sucrose use. To improve the efficiency of sucrose conversion in *C. beijerinckii*, we introduced a highly active sucrose invertase, encoded by *SUC2* from *S. cerevisiae*. This enzyme was effectively expressed in *C. beijerinckii*. Engineered strain CB1343 showed higher invertase activity than control strain CB1341 during both growth phases (table 3).

## 3.2. Effective sucrose transport for improved ABE fermentation

ABE fermentation was assessed using sucrose as the substrate with engineered strain CB1342 (expressing *StSUT1*) and control strain CB1341. Engineered strain CB1342 produced 7% more *n*-butanol (7.08 g l$^{-1}$) than the control strain (6.59 g l$^{-1}$) (table 4). The total solvent production in ABE fermentation by strain CB1342 was 11% higher than that of the control strain.

Sucrose transporter *StSUT1* belongs to the major facilitator superfamily and has the ability to significantly alter intracellular sucrose influx [29]. Additionally, a previous study indicated that sucrose PTS transport activity is repressed by fructose [19], but that *StSUT1* activity is not inhibited by substrates such as sucrose, fructose or glucose. From our previous study and the results shown in figure 2, it was evident that *StSUT1* induced high esculin transfer ability in *C. beijerinckii* and *C. acetobutylicum* strains [22]. Previous sucrose competition experiments demonstrated that sucrose

**Table 3.** Invertase activities of wild-type and recombinant strains. n.d., not detected.

| | invertase activity (U/L) | |
| strain | acidogenic phase | solventogenic phase |
|---|---|---|
| CB1341 | n.d. | n.d. |
| CB1342 | n.d. | n.d. |
| CB1343 | 22.6 ± 1 | 26.8 ± 3 |

**Table 4.** Fermentation parameters of recombinant strains in various media. FS, acetone–butanol–ethanol (ABE) fermentation medium with sucrose as the sole carbon source; FSJ, ABE fermentation medium with sugarcane juice as the feedstock; FSM, ABE fermentation medium with sugarcane molasses as the feedstock.

| medium | strain | residual sugar (g l$^{-1}$) | acetone (g l$^{-1}$) | butanol (g l$^{-1}$) | ethanol (g l$^{-1}$) | total solvent (g l$^{-1}$) | solvent yield (g g$^{-1}$ sucrose) |
|---|---|---|---|---|---|---|---|
| FS | CB1341 | 15.60 ± 0.37 | 1.44 ± 0.14 | 6.59 ± 0.41 | 0.23 ± 0.03 | 8.26 ± 0.59 | 0.34 ± 005 |
| | CB1342 | 11.32 ± 0.34 | 1.81 ± 0.00 | 7.08 ± 0.04 | 0.23 ± 0.01 | 9.13 ± 0.04 | 0.32 ± 0.02 |
| | CB1343 | 7.59 ± 3.85 | 2.11 ± 0.07 | 9.05 ± 0.88 | 0.30 ± 0.00 | 11.46 ± 0.95 | 0.36 ± 0.04 |
| FSJ | CB1341 | 18.59 ± 0.98 | 1.81 ± 0.03 | 7.13 ± 0.09 | 0.32 ± 0.00 | 9.27 ± 0.12 | 0.29 ± 0.02 |
| | CB1342 | 15.98 ± 0.1 | 1.95 ± 0.01 | 7.88 ± 0.01 | 0.32 ± 0.01 | 10.15 ± 0.01 | 0.30 ± 0.00 |
| | CB1343 | 12.73 ± 3.76 | 2.08 ± 0.13 | 8.88 ± 0.53 | 0.38 ± 0.02 | 11.34 ± 0.68 | 0.31 ± 0.04 |
| FSM | CB1341 | 21.57 ± 0.83 | 0.80 ± 0.01 | 4.19 ± 0.04 | 0.33 ± 0.02 | 5.32 ± 0.07 | 0.35 ± 0.01 |
| | CB1342 | 21.43 ± 1.68 | 0.94 ± 0.00 | 4.54 ± 0.08 | 0.35 ± 0.02 | 5.84 ± 0.06 | 0.39 ± 0.05 |
| | CB1343 | 16.51 ± 1.57 | 0.95 ± 0.04 | 6.82 ± 0.14 | 0.36 ± 0.00 | 8.13 ± 0.10 | 0.40 ± 0.02 |

competes with esculin for transport into recombinant strains [22]. The fermentation results in the present study confirmed that *StSUT1* induced sucrose transport in *C. beijerinckii*.

## 3.3. Enhanced sucrose use by co-expression of *StSUT1* and *SUC2*

*StSUT1* and *SUC2* were co-introduced into *C. beijerinckii* to investigate their effects on *n*-butanol production in the host. The resulting recombinant strain CB1343 produced 9.05 g l$^{-1}$ of *n*-butanol, which was 37% higher than that produced by control strain CB1341. Total ABE production of strain CB1343 was 11.46 g l$^{-1}$, which was 38.7% higher than that of the control strain. Strain CB1343 produced a solvent yield of 0.36 g g$^{-1}$ sucrose, which was 12.5% higher than that of strain CB1342, and 5.8% higher than that of control strain CB1341. Strain CB1343 consumed 32 g l$^{-1}$ sucrose, which was 33.3% and 14.8% higher than strains CB1341 and CB1342, respectively. These fermentation results support that introducing the heterologous pathway in *C. beijerinckii* improved sucrose consumption and enhanced ABE production (table 4).

A previously published method of improved sucrose fermentation in *Clostridium* by Zhang *et al.* [7] reported a 17.2% increase in ABE production. The recombinant strain overexpressed an endogenous sucrose usage pathway containing sucrose permease and sucrose phosphorylase, and demonstrated increased sucrose consumption and ABE production compared with the host strain [7]. The present study employed a different type of sucrose transporter mechanism. Our results are the first to demonstrate the effective application of a heterologous sucrose usage pathway, resulting in enhanced ABE production and sucrose consumption by engineered *Clostridium*.

## 3.4. ABE fermentation using sugarcane juice or molasses as feedstock

ABE fermentations were carried out with sugarcane juice or molasses as the feedstock. When sugarcane juice was used as the carbon source, engineered strain CB1342 produced 7.88 g l$^{-1}$ *n*-butanol with an

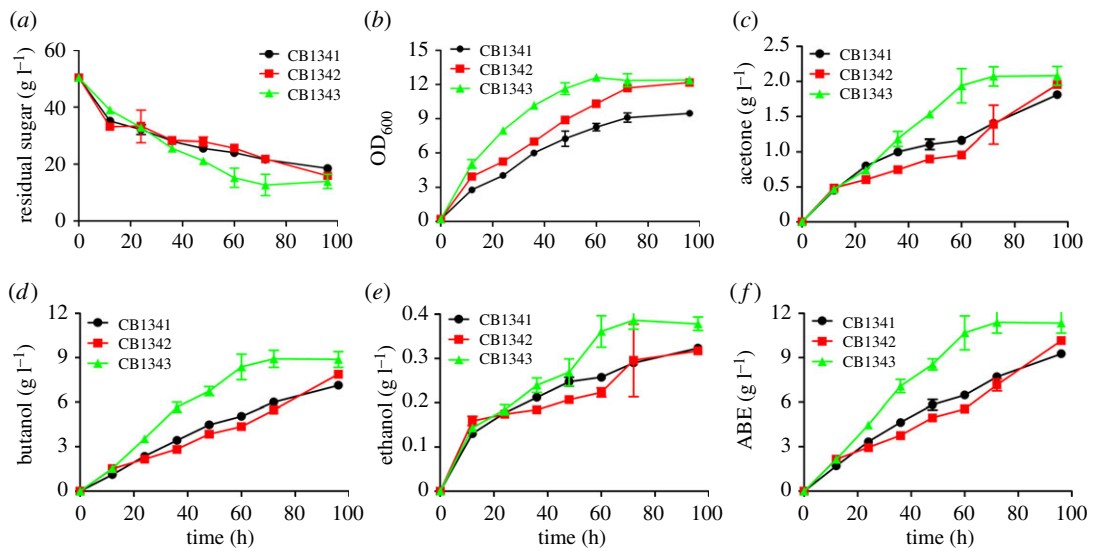

**Figure 3.** Acetone–butanol–ethanol (ABE) fermentation profiles of *C. beijerinckii* strain CB1341 (control), CB1342 (*C. beijerinckii* harbouring sucrose transporter *StSUT1*), and CB1343 (*C. beijerinckii* harbouring *StSUT1* and *SUC2*, a sucrose invertase). All strains were grown in fermentation medium with sugarcane juice as the feedstock. (*a*) Residual sugar; (*b*) biomass; (*c*) acetone production; (*d*) butanol production; (*e*) ethanol production; (*f*) ABE production.

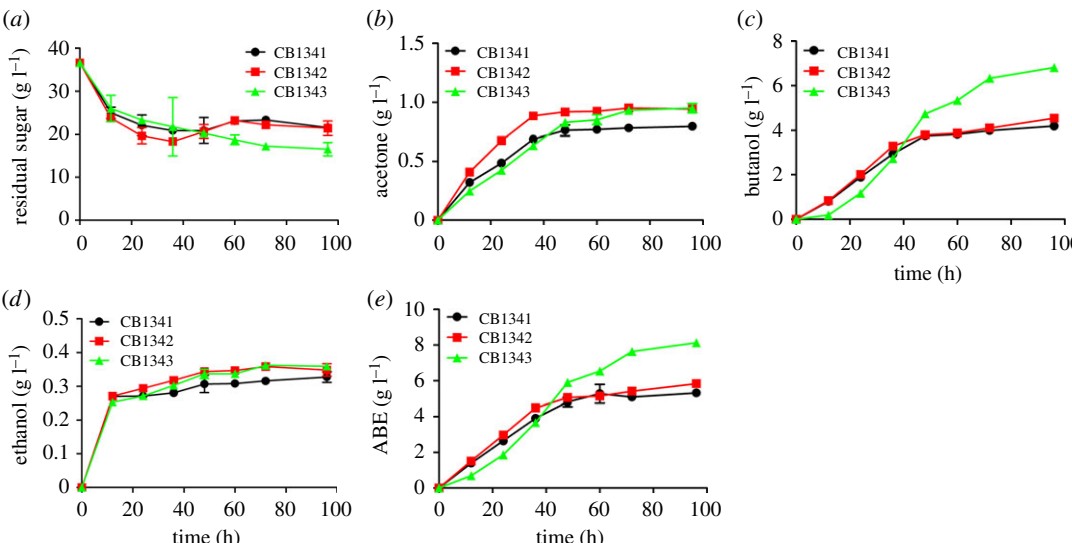

**Figure 4.** Acetone–butanol–ethanol (ABE) fermentation profiles of *C. beijerinckii* strain CB1341 (control), CB1342 (*C. beijerinckii* harbouring sucrose transporter *StSUT1*), and CB1343 (*C. beijerinckii* harbouring *StSUT1* and *SUC2*, a sucrose invertase). All strains were grown in fermentation medium with sugarcane molasses as the feedstock. (*a*) Residual sugar; (*b*) acetone production; (*c*) butanol production; (*d*) ethanol production; (*e*) ABE production.

*n*-butanol yield of 0.23 g g$^{-1}$ sucrose, which were 11.5% and 0.46% higher than those of the control strain CB1341, respectively. Engineered strain CB1343 produced 8.88 g l$^{-1}$ *n*-butanol with an *n*-butanol yield of 0.24 g g$^{-1}$ sucrose; the titre and yield were 24.5% and 11.3% higher than those of the control, respectively. The final ABE production reached 11.34 g l, which was 22.3% higher than that of the control (figure 3).

When sugarcane molasses was used as the carbon source, engineered strain CB1342 produced 4.54 g l$^{-1}$ *n*-butanol with an *n*-butanol yield of 0.30 g g$^{-1}$ sucrose, which were 8.4% and 0.83% higher than those of control strain CB1341, respectively. Engineered strain CB1343 produced 6.82 g l$^{-1}$ *n*-butanol with an *n*-butanol yield of 0.34 g g$^{-1}$ sucrose; the titer and yield were 62.8% and 21.9% higher than those of the control, respectively. The final ABE production reached 8.13 g l$^{-1}$, which was 53.1% higher than that of the control (figure 4). These results indicated that efficient sucrose transport

and metabolic pathways enhanced $n$-butanol production and yield when sugarcane molasses and juice were used as substrates.

# 4. Conclusion

In this study, sucrose utilization was increased by introducing a heterologous sucrose transport and metabolic pathway in *C. beijerinckii*. This pathway enhanced sucrose consumption and $n$-butanol production by *C. beijerinckii*. In particular, the ABE fermentation performance of the pathway co-expressing a heterologous sucrose transporter and invertase showed significant improvement when sucrose-based biomass (sugarcane molasses and juice) was used as the substrate. To our knowledge, this is the first study to achieve efficient sucrose metabolism in *C. beijerinckii* by directly introducing a heterologous sucrose transport and metabolic pathway. Similar strategies can be applied extensively in other microorganisms to increase the production of high-value biochemicals from sucrose or other inexpensive sucrose-based substrates.

Data accessibility. Data are deposited in the Zenodo (https://zenodo.org/record/5502692).

Authors' contributions. Conceptualization, H.P. and L.D.; investigation, Z. Z., L.L., H.T. and B.Z.; resources, Y.G. and Y.L.; writing—original draft preparation, L.L. and Z.Z.; writing—review and editing, H.P., R.H. and L.D.; funding acquisition, H.P. and L.D. All authors have read and agreed to the published version of the manuscript.

Competing interests. There are no conflicts to declare.

Funding. This research was funded by the Natural Science Foundation of China (grant no. 21566003), Basic operating expenses of Guangxi Academy of Sciences (grant no. 2019YJJ1001), the Natural Science Foundation of Guangxi (grant nos. 2020GXNSFAA259007, 2018GXNSFAA138103 and 2018GXNSFAA294047), and Guangxi Science and Technology Major Project (grant nos. AB17190534, 2018-15-Z03).

Acknowledgements. The authors are grateful to John M. Ward for gifting the *StSUT1* gene.

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
