## [Peer Review File · Royal Society Open Science]

Review History

RSOS-201858.R0 (Original submission)

Review form: Reviewer 1

Is the manuscript scientifically sound in its present form?

Yes

Are the interpretations and conclusions justified by the results?

Yes

Is the language acceptable?

Yes

Do you have any ethical concerns with this paper?

No

Have you any concerns about statistical analyses in this paper?

No

Recommendation?

Major revision is needed (please make suggestions in comments)

Comments to the Author(s)

In this study, a sucrose transporter gene from potato and an invertase gene from *Saccharomyces cerevisiae* were overexpressed in *Clostridium beijerinckii*. This improved the sucrose consumption and ABE production in the engineered strains. The work is interesting, and the results look promising. However, a lot of key information is missing in this manuscript, which must be addressed before the publication.

Comments:

1. what is the original source of the *C. beijerinckii* 13-4 strain? Is this isolated by the author, or is it obtained from a culture collection?
2. Also, why this strain (*C. beijerinckii* 13-4) is selected for this study, since there are so many other *Clostridium* strains (for example, the *C. beijerinckii* NCIMB 8052 strain) that perform much better for ABE production?
3. In Fig. 1 (C), it is indicated that the invertase (SUC2) was functional intracellularly. However, are there evidence to support this? Could it be possible that the invertase (or at least partially) be functional extracellularly?
4. For the expression of the genes (StSUT1 and SUC2), what promoter was used (for each of them)? Is codon optimization of these genes necessary for the expression since these genes are from plant and yeast, respectively.
5. Line 105 on page 5, and also in Table 4: the information about 'Residual sugar' from the fermentation is not so important. But the author should give the starting concentration of the sugars in the fermentation. Are the starting concentrations of sugars all the same for all the fermentations (especially when different carbon sources – sucrose, sugarcane juice, sugarcane molasses--are used)?

Review form: Reviewer 2**Is the manuscript scientifically sound in its present form?**

Yes

Are the interpretations and conclusions justified by the results?

No

Is the language acceptable?

No

Do you have any ethical concerns with this paper?

Yes

Have you any concerns about statistical analyses in this paper?

Yes

Recommendation?

Accept with minor revision (please list in comments)

Comments to the Author(s)

The manuscript stated the strategy to increase n butanol production via enhanced sucrose transport and Suc2 invertase activity.

I have one major question that require an explanation. The authors mentioned that (line 141) " the native sucrose hydrolase activity of *C. beijerinckii* is low" and the CB1342 displayed non-detectable invertase activity. How could you explain the n butanol produced in this strain? Plus, it is relatively high as compared to the CB1343 although comparable to CB1341.

How do you conclude on this difference? Significant or not?

Secondly, the language could be improved via editing service.

Also, If some statistical analysis, via annova such p value could be done and added, this will help to improve the confidentiality of data.

Decision letter (RSOS-201858.R0)

Dear Dr Lin

On behalf of the Editors, we are pleased to inform you that your Manuscript RSOS-201858 "Enhanced sucrose fermentation by introduction of heterologous sucrose transporter and invertase into *Clostridium beijerinckii* for ABE production" has been accepted for publication in Royal Society Open Science subject to minor revision in accordance with the referees' reports. Please find the referees' comments along with any feedback from the Editors below my signature.

Please submit your revised manuscript and required files (see below) no later than 7 days from today's (ie 07-Apr-2021) date. Note: the ScholarOne system will 'lock' if submission of the revision is attempted 7 or more days after the deadline. If you do not think you will be able to meet this deadline please contact the editorial office immediately.

Kind regards,

on behalf of Dr Nitin Baliga (Associate Editor) and Malcolm White (Subject Editor)
 openscience@royalsociety.org

Associate Editor Comments to Author (Dr Nitin Baliga):

Associate Editor: 1

Comments to the Author:

Please address concerns and revisions requested by both reviewers.

Reviewer comments to Author:

Reviewer: 1

Comments to the Author(s)

In this study, a sucrose transporter gene from potato and an invertase gene from *Saccharomyces cerevisiae* were overexpressed in *Clostridium beijerinckii*. This improved the sucrose consumption and ABE production in the engineered strains. The work is interesting, and the results look promising. However, a lot of key information is missing in this manuscript, which must be addressed before the publication.

Comments:

1. what is the original source of the *C. beijerinckii* 13-4 strain? Is this isolated by the author, or is it obtained from a culture collection?
2. Also, why this strain (*C. beijerinckii* 13-4) is selected for this study, since there are so many other *Clostridium* strains (for example, the *C. beijerinckii* NCIMB 8052 strain) that perform much better for ABE production?
3. In Fig. 1 (C), it is indicated that the invertase (SUC2) was functional intracellularly. However, are there evidence to support this? Could it be possible that the invertase (or at least partially) be functional extracellularly?
4. For the expression of the genes (StSUT1 and SUC2), what promoter was used (for each of them)? Is codon optimization of these genes necessary for the expression since these genes are from plant and yeast, respectively.
5. Line 105 on page 5, and also in Table 4: the information about 'Residual sugar' from the fermentation is not so important. But the author should give the starting concentration of the sugars in the fermentation. Are the starting concentrations of sugars all the same for all the fermentations (especially when different carbon sources – sucrose, sugarcane juice, sugarcane molasses--are used)?

Reviewer: 2

Comments to the Author(s)

The manuscript stated the strategy to increase n butanol production via enhanced sucrose transport and Suc2 invertase activity.

I have one major question that require an explanation. The authors mentioned that (line 141) " the native sucrose hydrolase activity of *C. beijerinckii* is low" and the CB1342 displayed non-detectable invertase activity. How could you explain the n butanol produced in this strain? Plus, it is relatively high as compared to the CB1343 although comparable to CB1341.

How do you conclude on this difference? Significant or not?

Secondly, the language could be improved via editing service.

Also, If some statistical analysis, via annova such p value could be done and added, this will help to improve the confidentiality of data.

===PREPARING YOUR MANUSCRIPT===

===PREPARING YOUR REVISION IN SCHOLARONE===

- 1) One version identifying all the changes that have been made (for instance, in coloured highlight, in bold text, or tracked changes);
 - 2) A 'clean' version of the new manuscript that incorporates the changes made, but does not highlight them.
 - An individual file of each figure (EPS or print-quality PDF preferred [either format should be produced directly from original creation package], or original software format).
 - An editable file of each table (.doc, .docx, .xls, .xlsx, or .csv).
 - An editable file of all figure and table captions.
- Note: you may upload the figure, table, and caption files in a single Zip folder.
- Any electronic supplementary material (ESM).
 - If you are requesting a discretionary waiver for the article processing charge, the waiver form must be included at this step.
 - If you are providing image files for potential cover images, please upload these at this step, and inform the editorial office you have done so. You must hold the copyright to any image provided.
 - A copy of your point-by-point response to referees and Editors. This will expedite the preparation of your proof.

- Ensure that your data access statement meets the requirements at <https://royalsociety.org/journals/authors/author-guidelines/#data>. You should ensure that you cite the dataset in your reference list. If you have deposited data etc in the Dryad repository, please only include the 'For publication' link at this stage. You should remove the 'For review' link.
- If you are requesting an article processing charge waiver, you must select the relevant waiver option (if requesting a discretionary waiver, the form should have been uploaded at Step 3 'File upload' above).
- If you have uploaded ESM files, please ensure you follow the guidance at <https://royalsociety.org/journals/authors/author-guidelines/#supplementary-material> to include a suitable title and informative caption. An example of appropriate titling and captioning may be found at [https://figshare.com/articles/Table_S2_from_Is_there_a_trade-off_between_peak_performance_and_performance_breadth_across_temperatures_for_aerobic_sc ope_in_teleost_fishes_/3843624](https://figshare.com/articles/Table_S2_from_Is_there_a_trade-off_between_peak_performance_and_performance_breadth_across_temperatures_for_aerobic_scope_in_teleost_fishes_/3843624).

Author's Response to Decision Letter for (RSOS-201858.R0)

See Appendix A.

RSOS-201858.R1 (Revision)

Review form: Reviewer 1

Is the manuscript scientifically sound in its present form?

No

Are the interpretations and conclusions justified by the results?

Yes

Is the language acceptable?

Yes

Do you have any ethical concerns with this paper?

No

Have you any concerns about statistical analyses in this paper?

No

Recommendation?

Accept with minor revision (please list in comments)

Comments to the Author(s)

1. Since *C. beijerinckii* 13-4 strain was isolated by the authors, and this is the first time for them to report this strain, they should give more background information about this strain. Where is the strain isolated from? How is the strain isolated? What are the primary features of this strain, especially compared to other solventogenic clostridial strains, especially to the type strain *Clostridium beijerinckii* NCIMB 8052.
2. Related to 1, I was still not convinced why *C. beijerinckii* 13-4 strain is selected for this study, since the authors did not demonstrate the advantages of this strain over *C. beijerinckii* NCIMB 8052 or *C. acetobutylicum* DSM 792.

Decision letter (RSOS-201858.R1)

Dear Dr Lin

On behalf of the Editors, we are pleased to inform you that your Manuscript RSOS-201858.R1 "Enhanced sucrose fermentation by introduction of heterologous sucrose transporter and invertase into *Clostridium beijerinckii* for ABE production" has been accepted for publication in Royal Society Open Science subject to minor revision in accordance with the referees' reports. Please find the referees' comments along with any feedback from the Editors below my signature.

Please submit your revised manuscript and required files (see below) no later than 7 days from today's (ie 23-Jul-2021) date. Note: the ScholarOne system will 'lock' if submission of the revision is attempted 7 or more days after the deadline. If you do not think you will be able to meet this deadline please contact the editorial office immediately.

on behalf of Dr Nitin Baliga (Associate Editor) and Malcolm White (Subject Editor)
openscience@royalsociety.org

Associate Editor Comments to Author (Dr Nitin Baliga):

The reviewer has made two minor but important requests (see their review). Please provide details on the strain and why it was selected.

Reviewer comments to Author:

Reviewer: 1

Comments to the Author(s)

1. Since *C. beijerinckii* 13-4 strain was isolated by the authors, and this is the first time for them to report this strain, they should give more background information about this strain. Where is the strain isolated from? How is the strain isolated? What are the primary features of this strain, especially compared to other solventogenic clostridial strains, especially to the type strain *Clostridium beijerinckii* NCIMB 8052.
2. Related to 1, I was still not convinced why *C. beijerinckii* 13-4 strain is selected for this study, since the authors did not demonstrate the advantages of this strain over *C. beijerinckii* NCIMB 8052 or *C. acetobutylicum* DSM 792.

===PREPARING YOUR MANUSCRIPT===

Your revised paper should include the changes requested by the referees and Editors of your manuscript. You should provide two versions of this manuscript and both versions must be provided in an editable format:
one version identifying all the changes that have been made (for instance, in coloured highlight, in bold text, or tracked changes);
a 'clean' version of the new manuscript that incorporates the changes made, but does not highlight them. This version will be used for typesetting.

===PREPARING YOUR REVISION IN SCHOLARONE===

-- Ensure that your data access statement meets the requirements at <https://royalsociety.org/journals/authors/author-guidelines/#data>. You should ensure that you cite the dataset in your reference list. If you have deposited data etc in the Dryad repository, please only include the 'For publication' link at this stage. You should remove the 'For review' link.

Author's Response to Decision Letter for (RSOS-201858.R1)

See Appendix B.

Decision letter (RSOS-201858.R2)

Dear Dr Lin,

I am pleased to inform you that your manuscript entitled "Enhanced sucrose fermentation by introduction of heterologous sucrose transporter and invertase into *Clostridium beijerinckii* for ABE production" is now accepted for publication in Royal Society Open Science.

on behalf of Dr Nitin Baliga (Associate Editor) and Malcolm White (Subject Editor)
openscience@royalsociety.org

Appendix A

point-by-point response

Reviewer: 1

1. what is the original source of the *C. beijerinckii* 13-4 strain? Is this isolated by the author, or is it obtained from a culture collection?

A: The *C. beijerinckii* 13-4 strain was isolated by our laboratory. This information was added that highlighted in red in the table 1.

2. Also, why this strain (*C. beijerinckii* 13-4) is selected for this study, since there are so many other Clostridium strains (for example, the *C. beijerinckii* NCIMB 8052 strain) that perform much better for ABE production?

A: *C. beijerinckii* 13-4 is a fermentation strain with sucrose as substrate. This strain was isolated by our lab and was used as fermentation stain for butanol production.

In our previous paper (Molecules, 2019,24(19):3495), we have showed that transporter SUT1 can function in model strains *C. beijerinckii* NCIMB 8052 and *C. acetobutylicum* DSM 792. These results indicated the universality of transport function of SUTI in *C. beijerinckii*. In this work, fermentation study was performed, and results confirmed that SUT1 did contribute to the fermentation of *clostridium*.

3. In Fig. 1 (C), it is indicated that the invertase (SUC2) was functional intracellularly. However, are there evidence to support this? Could it be possible that the invertase (or at least partially) be functional extracellularly?

A: In this work, enzyme SUC2 was designed to functioned intracellularly. SUC2 gene without signal peptide was cloned to plasmid. Thus, this enzyme was expressed intracellular.

In theory, it is possible for some SUC2 to leak out of the cell during cell autolysis.

We do test the SUC2 enzyme activity under different occasions. To measure enzyme activity, cells from the acidogenesis phase (24 h) and solventogenesis phase (48 h) were harvested by centrifugation. The cells were disrupted using a sonication device to obtain cell lysates. The lysate was then centrifuged at 12,000 g for 20 min at 4 °C, and the supernatant was collected to measure enzyme activity. The test showed that the SUC2 enzyme activity or invertase activity of this recombinant strain was functioned intracellularly in the test occasions.

4. For the expression of the genes (*StSUT1* and *SUC2*), what promoter was used (for each of them)? Is codon optimization of these genes necessary for the expression since these genes are from plant and yeast, respectively.

A: *StSUT1* and *SUC2* were constructed as a polycistron under the thiolase promoter of the pSOS95 plasmid. There is no codon optimization for either gene. This content is added to the 94- 96 lines of the manuscript and is marked in red.

5. Line 105 on page 5, and also in Table 4: the information about 'Residual sugar' from the

fermentation is not so important. But the author should give the starting concentration of the sugars in the fermentation. Are the starting concentrations of sugars all the same for all the fermentations (especially when different carbon sources—sucrose, sugarcane juice, sugarcane molasses—are used)?

A: The starting concentrations of sugars was given in the text of Line 130 on page 4, the text is the starting concentration of the sugars---‘A sucrose medium was used as the ABE fermentation medium, containing 40 g/L sucrose . Sugarcane molasses (containing 310 g/L sucrose, 52 g/L glucose, and 49 g/L fructose) or sugarcane juice (containing 100 g/L sucrose, 29 g/L glucose, and 13 g/L fructose) was diluted and used to replace sucrose in the ABE fermentation medium.’.

The same carbon sources had the same starting concentrations for all the fermentations. Different carbon sources had different sugar concentrations as stated in the text.

Reviewer: 2

1.The authors mentioned that (line 141) " the native sucrose hydrolase activity of *C. beijerinckii* is low" and the CB1342 displayed non-detectable invertase activity. How could you explain the n butanol produced in this strain?

A: For *C. beijerinckii*, there are native sucrose phosphotransferase system (sucrose-PTS), which transfer sucrose into the cell and produce sucrose-6-P within the cell. Thus, *C. beijerinckii* can ferment sucrose into butanol.

2.Plus, it is relatively high as compared to the CB1343 although comparable to CB1341. How do you conclude on this difference? Significant or not?

A: Through the introduction of sucrose transporter and invertase enzyme, the recombinant strain had remarkable sucrose fermentation ability. It is significant.

3.Secondly, the language could be improved via editing service.

A: The manuscript has been edited for proper English language, grammar, punctuation, spelling, and overall style by the highly qualified native English speaking editors at Wiley Editing Services. We can show the Wiley Editing Services certificate.

4.Also, If some statistical analysis, via annova such p value could be done and added, this will help to improve the confidentiality of data.

A: All fermentations data were performed in duplicate and calculate the standard deviation. Though annova statistical analysis is a good analysis method for biology experiment, but we believe that standard deviation is sufficient for the comparison of bacterial fermentations.

Appendix B

point-by-point response

1. Since *C. beijerinckii* 13-4 strain was isolated by the authors, and this is the first time for them to report this strain, they should give more background information about this strain. Where is the strain isolated from? How is the strain isolated? What are the primary features of this strain, especially compared to other solventogenic clostridial strains, especially to the type strain *Clostridium beijerinckii* NCIMB 8052.

A: The isolating and screening method is given in section of *Microorganisms and cultivation condition* in the revised manuscript and is marked in red.

A newly provided explanatory text was given in section of *Heterologous expression of StSUT1 and SUC2 in C. beijerinckii*. and is marked in red.

2. Related to 1, I was still not convinced why *C. beijerinckii* 13-4 strain is selected for this study, since the authors did not demonstrate the advantages of this strain over *C. beijerinckii* NCIMB 8052 or *C. acetobutylicum* DSM 792.

A: As for ABE production with sucrose as feedstock, it is well known that *C. beijerinckii* is superior to *C. acetobutylicum*. In fact, *C. acetobutylicum* hardly uses sucrose as raw material for butanol fermentation, although it can use glucose well.

Strain 13-4 is a strain suitable for industrial fermentation. Compared with 8052, the strain can be grown and fermented in a simple medium. Such characteristics are desirable for industrial strains.

Indeed, the ABE fermentation yield of strain 13-4 was low, while the reported high yield of strain BA101 was 17.7 g/L.

But the purpose of this paper is to demonstrate the possibility of introducing heterogeneous sucrose transporters to help clostridium fermentation. It is proved that even industrial candidate strain can be modified to achieve this capability. Further studies are needed to improve the fermentation capacity of this recombinant strain to help or improve sucrose metabolism on the basis of high sugar intake.